



# Soil health approaches to assess the impacts of no-tillage with agricultural terraces in southern Brazil

Ariane Lentice de Paula[1], Luis Miguel Schiebelbein[1], Regiane Kazmierczak Becker[1], Eduardo Augusto Agnellos Barbosa[1], Fabrício Tondello Barbosa[1], Carolina Weigert Galvão[1], Rafael Mazer Etto[1],
Heverton Fernando Melo[1], Adriel Ferreira da Fonseca[1], Neyde Fabiola Balarezo Giarola[1]

[1]Department of Soil Science and Agricultural Engineering, State University of Ponta Grossa, Ponta Grossa, Paraná, Brazil

*Correspondence to*: Ariane Lentice de Paula (ariane.le.paula@gmail.com)

**Abstract.** Soil health assessment depends on the appropriate selection of indicators and robust, sensitive methods for its determination. In this study, four integrative approaches were evaluated to assess the impacts of no-till with and without
agricultural terraces on soil health in Southern Brazil. The different methods used were: (1) Principal Component Analysis (PCA); (2) expert opinion (EO); (2) FERTBIO; and (4) Soil Management Assessment Framework (SMAF). All approaches followed four steps: (i) selection of indicators; (ii) interpretation of indicators; (iii) integration of indicators; and (iv) calculation of soil health indices. The methods varied in the steps of indicator selection, interpretation, and the method of indicator integration. The indicators used included physical (bulk density, total porosity, soil penetration resistance, and
water retention capacity), chemical (pH, calcium, phosphorus, potassium, organic matter, CEC, and base saturation), and biological indicators (microbial biomass carbon, β-glucosidase, and arylsulfatase). Crop yield was evaluated for maize (2019/20 and 2021/22 harvests), wheat (2021 harvest), and soybean (2020/21 harvest). Descriptive statistics, median comparisons, principal component analysis and spearman correlation analysis were applied for the analysis of results. The results showed that only the EO and FERTBIO approaches were sensitive enough to detect differences in soil health between
management systems, indicating that no-till with terraces had better soil health. Biological indicators were more sensitive in differentiating treatments, showing a rapid response in the short term. Maize (2019/20 harvest) and wheat (2021 harvest) yields were higher under the no-till with terraces treatment. Over time, yield showed a stronger relationship with soil health. The results highlight the importance of selecting appropriate indicators for soil health assessment and reinforce the benefits of agricultural terracing for the sustainability of production systems.

**1 Introduction**

Soil is the foundation of terrestrial ecosystems, playing a crucial role in agricultural sustainability, the maintenance of natural resources, and environmental quality (Doran and Zeiss, 2000; Hartmann et al., 2015). Its integrity determines the ability to provide essential nutrients to plants, sustain food chains, and promote ecosystem stability (Lal, 2009). Healthy and biologically active soil not only support agricultural productivity but also contribute to environmental resilience and global





food security (FAO, 2015). Soil health refers to its functional capacity as a dynamic and living ecosystem that supports plant and animal productivity, regulates water and air quality, and contributes to environmental stability (Doran et al., 1996). However, this condition is directly influenced by management practices and can be preserved or degraded based on human interventions (USDA, 2023). Maintaining soil health aligns with various UN Sustainable Development Goals and the One Health concept (Banerjee and Van der Heijden, 2023).

Soil health assessment involves multiple interactions among physical, chemical, and biological components. Approaches that generate indices sensitive to soil use and management modifications are employed to quantify its functionality and sustainability (Andrews et al., 2004; Bünemann et al., 2018; Doran and Zeiss, 2000; Simon et al., 2022). Systematic measurements of these indicators support strategies to conserve productivity and environmental quality (Andrews and Carroll, 2001; Obade et al., 2014).

Soil health approaches can generate specific or integrated indices. Specific indices assess physical, chemical, and biological attributes separately. While useful for understanding individual soil aspects, specific indices may not fully capture interactions between components. For a broader approach, tests combining multiple indicators generate integrated soil health indices, allowing for a holistic evaluation of soil conditions and agricultural sustainability (Cherubin et al., 2016a; Lima et al., 2024; Luz et al., 2019).

The Soil Management Assessment Framework (SMAF) is an example of a test generating an integrated index capable of providing a quantitative and standardized soil health assessment across agricultural contexts (Andrews et al., 2004). Its effectiveness in comparing management systems and identifying practices that promote sustainability has encouraged its use (Becker et al., 2025, 2024; Cherubin et al., 2021, 2017, 2016b; Jimenez et al., 2022; Lima et al., 2024; Lisboa et al., 2019; Luz et al., 2019; Matos et al., 2022; Pereira et al., 2024; Ruiz et al., 2020). In Brazil, FERTBIO was developed to integrate

soil fertility and biology, considering interactions between nutrients and microbial activity, offering a more accurate assessment of soil conditions in tropical regions. Studies show that FERTBIO is effective in detecting changes in soil health caused by different agricultural management practices (Mendes et al., 2021a, 2021b).

Other approaches used to generate integrated soil health indices include Principal Component Analysis (PCA) and expert opinion (EO). PCA is a statistical technique that reduces data dimensionality and identifies the main factors contributing to

soil quality variability (Andrews et al., 2002; Marion et al., 2022; Shukla et al., 2006). EO relies on technical knowledge and experience to select and weigh indicators, allowing for an evaluation tailored to specific management and environmental conditions (Andrews et al., 2004; Barrios, 2007).

Conservation systems, compared to conventional agriculture, promote better soil conditions, leading to increased carbon sequestration, better water retention capacity, and a more active and diverse microbial community (Gattinger et al., 2012;

Gomiero et al., 2011; Lori et al., 2017; Tuck et al., 2014). In Brazil, no-till is widely adopted to preserve soil structure, improve moisture retention, and reduce erosion (Derpsch et al., 2010; Possamai et al., 2022; Telles et al., 2019; Wen et al., 2023). However, soil erosion is an ongoing challenge, especially in areas with rugged terrain. Conservation techniques such as no-till and terracing have been essential strategies to mitigate soil degradation and improve agricultural sustainability in



the region (Merten et al., 2015). Terracing combined with no-till can reduce soil erosion and enhance water infiltration, but
its effects on soil health in tropical agriculture remain unclear (Fuentes-Guevara et al., 2024; Lal, 2020; Panagos et al.,
2015).

Given Brazil's leadership in tropical agriculture and the widespread adoption of no-tillage, which is applied to 33 million
hectares of the country's cropland (Fuentes-Llanillo et al., 2021), but which alone is not sufficient to control soil loss on
sloping soils (Merten et al. 2015), the aim of this study was to evaluate the effects of agricultural terracing under no-till
systems over three years, using four different integrative approaches to assess soil health. Additionally, we seek to identify
the most robust and sensitive SHI approaches and soil indicators for detecting short-term changes in tropical soil health.

## 2 Material and methods

### 2.1 Location and characterization of the experimental site

The experimental field belongs to the State University of Ponta Grossa (UEPG) and is located in Ponta Grossa, Paraná,
Brazil, at 25°05'49"S, 50°02'42"W, with an altitude of 1,015 meters (Fig. 1). The predominant climate is Cfb (Köppen), a
humid subtropical climate with mild summers, no dry season, and frequent, severe frosts (Alvares et al., 2013). The soil in
the study area was classified as a Latossolo Vermelho-Amarelo distrófico (Santos et al., 2018) or Dystrophic Typic
Hapludox (Soil Survey Staff, 2014), with a sandy clay loam texture (Table 1).

In the 1980s, conventional tillage predominated in the study area for grain cultivation, employing mechanical conservation
practices (terraces) to control erosion and conserve water. In the early 1990s, it transitioned to no-till with occasional
subsoiling, and the terraces were removed. In 2018, this experiment was initiated with the surface application of 2.3 t ha⁻¹ of
limestone, followed by soil subsoiling to a depth of 0.3 m. Finally, the experiment was set up with two mega-plots, both
maintained under no-till: (1) no-till (NT) and (2) no-till with terraces (NT+T). In each mega-plot, regular grids of 36
georeferenced points (n=36) were defined (Fig. 1) for soil sample collection. Soil sampling was carried out between May and
June in the years 2020, 2021, and 2022.



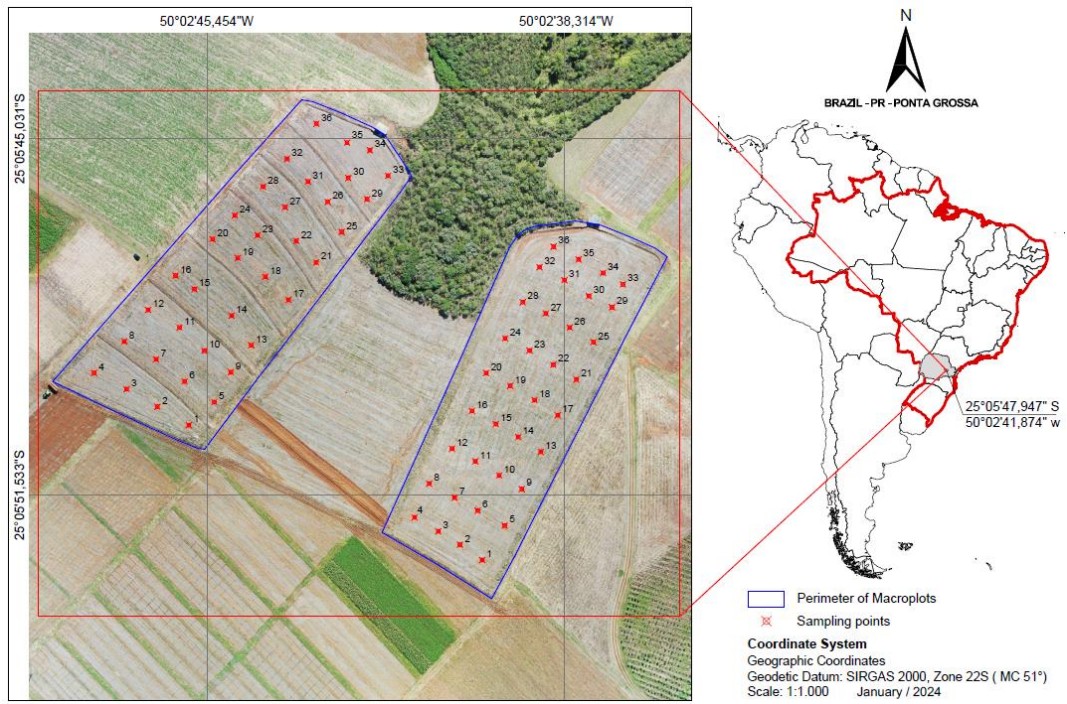

**Figure 1. Image of the experimental plots and the regular grid of soil sample collection points (Silva, 2025). On the left, a megaplot cultivated under no-till with terraces (NT+T); on the right, a mega-plot cultivated under no-till (NT).**

90 **Table 1.** Soil granulometry of the experimental mega-plots

| Soil management | Sand | Silt | Clay | Soil texture |
|:---:|:---:|:---:|:---:|:---:|
| | | (g kg$^{-1}$) | | |
| **NT** | 610 | 92 | 298 | Sandy clay loam |
| **NT+T** | 643 | 78 | 279 | Sandy clay loam |

NT = No-till; NT+T = No-till with terraces; average values of 36 sample points.

Annual cycle species were cultivated according to the most common management and sequence in the region: soybean (Glycine max) and corn (Zea mays) in the summer, and wheat (Triticum aestivum) and black oat (Avena strigosa) in the
95 winter.

**2.2 Soil properties analyses**

Twenty-six soil indicators were used to compose the different SHIs over the years 2020, 2021, and 2022. The methods used for the determination of soil indicators: bulk density (BD), total porosity (TP), macroporosity (Macro), microporosity (Micro), field capacity (FC), permanent wilting point (PWP), plant available water (PAW), soil water storage capacity
100 (SWSC), soil aeration capacity (SAC), soil penetration resistance (SPR), pHH20, pHCaCl$_2$, aluminum (Al), calcium (Ca),





magnesium (Mg), potassium (K), available phosphorus (P), cation exchange capacity (CEC), cation exchange capacity at pH 7.0 (CECpH7.0), sum of bases (SB), base saturation (BS), soil organic carbon (OC), soil organic matter (SOM), microbial biomass carbon (MBC), beta-glucosidase enzyme activity (BG), arylsulfatase enzyme activity (A) are presented in Supplementary Table 1.

**2.3 Procedures for determining Soil Health Indices (SHIs)**

The tests used to determine the SHIs followed three steps: (i) indicator selection; (ii) indicator interpretation; and (iii) indicator integration (Cherubin et al., 2016a) (Fig. 2). The specifics of each step using the different tests are detailed below.

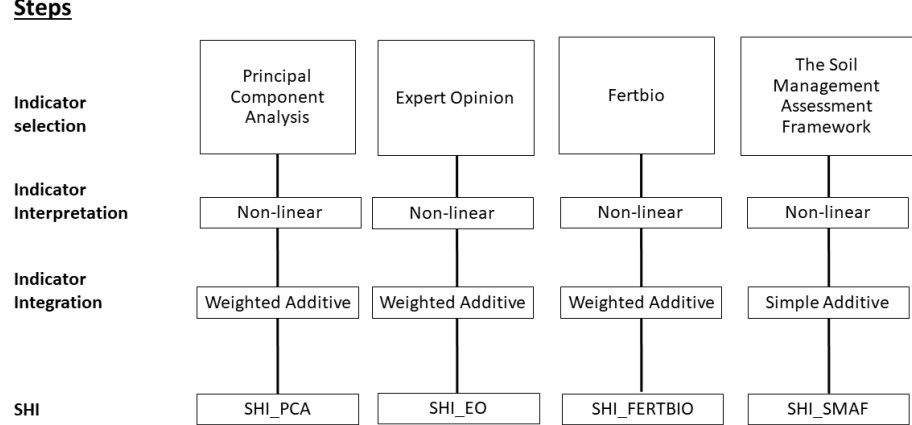

**Figure 2. Steps followed to generate soil health indices: SHI_PCA, SHI_EO, SHI_FERTBIO and SHI_SMAF.**

**2.3.1 Step 1: Indicator selection**

PCA was used to mitigate subjectivity in the selection of indicators, with the aid of the FactoMineR package (Lê et al., 2008). Principal components (PCs) with eigenvalues >1% were considered, following Kaiser's criteria. Within each PC, only variables with high factor loadings were selected (with 10% as the absolute value of the factor loading). In cases where multiple variables were selected within a PC, a Spearman correlation analysis (r ≥ 0.80) was performed to identify and eliminate redundant variables in determining the SHI (Gu et al., 2019). If high correlation between indicators was verified, the indicator with the highest weight within the PC was prioritized.

In the EO, the selection is based on expert opinion and literature. This selection prioritizes the indicators that are most sensitive to variations in soil behavior and those that are easiest to determine in the field or laboratory and to interpret. In the case of FERTBIO, the indicators are those proposed by Mendes et al. (2021b). For the SMAF, the indicators that best represent soil functions proposed by (Andrews et al., 2004; Stott et al., 2010; Wienhold et al., 2009) were selected.



### 2.3.2 Step 2: Indicator interpretation

For the SHI_PCA and SHI_EO indices, the determination was conducted according to the methodology proposed by Karlen and Stott (1994). All selected indicators were transformed and standardized on scales ranging from 0 to 1, using the non-linear score standardization equation, as described by Wymore (1993):


$$\text{Score} = \frac{1}{1+\left(\frac{B-L}{x-L}\right)^{2S(B+x-2L)}} \tag{1}$$

where, Score: standardized value of the indicator; B: critical value of the indicator with a score equal to 0.5; L: lower limit of the indicator; S: slope of the tangent of the curve; x: measured value of the indicator. The limits are presented in Table 2. According to Karlen and Stott, (1994), soil indicators can be standardized using "more is better," "optimal value," and "less is better" curves (Figure S1). For each curve, the slope of the tangent was calculated as described in (Wymore, 1993).


**Table 2.** Limits for soil health indicators and scoring curves

| Soil Indicator | Unit | Lower Threshold | Lower Baseline | Upper Threshold | Upper Threshold | Optimum point | Scoring curve | Reference |
|---|---|---|---|---|---|---|---|---|
| | | | | **Physical** | | | | |
| BD | Mg dm$^{-3}$ | 1.30 | 1.45 | 1.65 | | | Less is better | (Reichert et al., 2003) |
| TP | m³ m$^{-3}$ | 0.32 | 0.47 | 0.61 | | | More is better | (Libardi, 2012) |
| SPR | MPa | 2 | 3 | 5 | | | Less is better | (Arshad et al., 1997) |
| SWSC | - | 0.38 | 0.50 | 0.92 | 0,80 | 0.60-0.70 | Optimum | (Reynolds et al., 2007) |
| | | | | **Chemical** | | | | |
| pH$_{CaCl2}$ | - | 4 | 4.5 | 6.5 | 7 | 5.5 | Optimum | (Pavinato et al., 2017) |
| P | mg dm$^{-3}$ | 4 | 12 | 18 | | | More is better | (Pavinato et al., 2017) |
| K | cmol$_c$ dm$^{-3}$ | 0.06 | 0.21 | 0.45 | | | More is better | (Pavinato et al., 2017) |
| Ca | cmol$_c$ dm$^{-3}$ | 0.5 | 2 | 6 | | | More is better | (Pavinato et al., 2017) |
| CECpH7.0 | cmol$_c$ dm$^{-3}$ | 5 | 14 | 24 | | | More is better | (Pavinato et al., 2017) |
| BS | % | 20 | 50 | 70 | | | More is better | (Pavinato et al., 2017) |
| | | | | **Biological** | | | | |
| OC | g dm$^{-3}$ | 4 | 14 | 20 | | | More is better | (Pavinato et al., 2017) |



| | | | | | | | |
|---|---|---|---|---|---|---|---|
| **MBC** | mg kg⁻¹ | 200 | 275 | 350 | | More is better | (Lopes et al., 2013) |
| **BG** | mg *p*-nitrophenol kg⁻¹ soil h⁻¹ | 60 | 90 | 120 | | More is better | (Lopes et al., 2013) |
| **A** | mg *p*-nitrophenol kg⁻¹ soil h⁻¹ | 38 | 60 | 90 | | More is better | (Lopes et al., 2013) |

BD: Bulk Density; TP: Total Porosity; SPR: Soil Penetration Resistance; SWSC: Soil Water Storage Capacity; P: Available Phosphorous; K: Exchangeable Potassium; Ca: Exchangeable Calcium; $CEC_{pH7.0}$: Cation Exchange Capacity at pH 7.0; BS: Base saturation; OC: Organic Carbon; MBC: Microbial Carbon Biomass; BG: Beta-glucosidase Activity.


For SHI_FERTBIO, the interpretation of the indicators was also based on Karlen and Stott (1994) and on the equations of Wymore (1993). However, the calculations were performed using the Soil Quality Interpretation Module (MIQS) contained in the BioAS technology (Mendes et al., 2021b). For SHI_SMAF, the indicators were standardized according to the algorithms previously published (Andrews et al., 2004; Stott et al., 2010; Wienhold et al., 2009). (Table S2).

**2.3.3 Step 3: Indicator integration**

For SHI_SMAF, the integration was done using the simple additive method. For the others, the integration was done using the weighted additive method, varying the weight factor. SHI_PCA was integrated using the eigenvalue weights; in SHI_FERTBIO, the process followed as described by Mendes et al. (2021b).

The SHI_EO was integrated using the covariance matrix weights, in a manner based on that described in Manly (2008) and 145 Jolliffe (2002) applied to different multivariate analysis methods. The SHI_EO integration method consisted of an analysis of the proportional contribution of variables to a composite index. The index I is equivalent to the sum of the individual values of the variables ($I = \sum_{i=1}^{k} w_i x_i$) for each point, where in this case the value of wi is equal to one for all the variables and the value of xi is variable from 0 to 1, the scale effect of the variables can be disregarded. Thus, the index I can be considered as a linear function of the individual indicators of the variables used, and covariance is one of the ways of 150 assessing the association between the variations observed in the I-index and the variations observed in the variables used to estimate it.

From this, we calculated the covariance between the values of and I, ($Cov(x_{ij}, I) = \frac{1}{n}\sum_{j=1}^{n}(x_{ij} - \bar{x_i})(I_j - \bar{I})$) where xij and Ij are the values of the variables xi and I to the point j, deducted from their respective mean values to obtain the total joint variability, which measures the total contribution of the xi variables to the total variation of the I index. As the covariance is 155 linear, the sum reflects the combined joint variation of the variables with the I index.



The coefficient of proportionality (pi) for each variable is given by ($pi = \frac{Cov(x_{ij}, I)}{\sum_{j=1}^{k} Cov(x_j, I)}$), which reflects the contribution of variable xi to the total variation of index I, and also that $\sum p_i = 1$. Thus, the SHI-EO value for each point was calculated from ($SHI\_EO = \sum_{i-1}^{k} p_i . I$).

## 2.4 Crop yields

Sampling for yield determination of maize (2019/20 and 2021/22 harvests), wheat (2021 harvest), and soybean (2020/21 harvest) was carried out at 36 sampling points, following the methodology described by Bach et al. (2020).

## 2.5 Statistical analyses

Statistical differences were analyzed using the Wilcoxon test ($p < 0.01$, $p < 0.05$ and $p < 0.10$). Principal Component Analysis (PCA) was used to select indicators, as well as to evaluate the relationship between SHI scores and productivity ($p \leq$

0.05). Spearman correlation analysis was used to reduce the number of indicators after their initial selection by PCA. The comparison among the SHIs was performed using the Kruskal-Wallis test, followed by the Dunn test for multiple comparisons. All statistical analyses were conducted in R software, version 4.2.3.(R Development Core Team, 2023).

## 3 Results

### 3.1 Soil Indicators

The effects of agricultural terracing were assessed by monitoring individual indicators over multiple years (Table S3). In 2020, the NT+T treatment exhibited higher values of SPR, SWSC, K, MBC, and BG, and lower levels of P, Ca, CECs, OC, SOM, and A compared to NT. In 2021, NT+T continued to show elevated SPR, SWSC, and K, along with reduced P and A. Additionally, Mg increased and pHCaCl₂ decreased relative to NT. That year, CEC also became higher and BG lower in NT+T than in NT. By 2022, SPR remained consistently higher and pHCaCl₂ lower in NT+T, as observed in previous years.

Notably, OC, SOM, and A—which were lower in NT+T in 2020—became higher after two years. In contrast, MBC and BG, initially higher in NT+T, showed lower values than NT by 2022.

### 3.2 Selection Indicators

Given the variability in indicator behavior, and to better assess the effects of terracing along the time, the indicators were integrated into four soil heath indices, each with specific criteria for selection, interpretation, and integration (Fig. 2). For

SHI_PCA, principal component analysis resulted in the selection of 5 principal components, which explained 71.69% of the variability in the data (Table S4). After Spearman's correlation (Figure S2), considering the correlation within each principal component ($r \geq 0.80$), the data set was reduced to 8 indicators: a) physical: BD, TP, SWSC; b) chemical: Ca, P, CECpH7.0, BS; c) biological: MBC. For SHI_EO, nine indicators were selected, three from each group: physical (BD, SPR, and SWSC),





chemical (pHCaCl₂, P, and K), and biological (OC, BG, and A). For SHI_FERTBIO, the index is composed of chemical

(FERT) and biological (BIO) indicators represented by three soil functions: soil capacity to recycle nutrients (BG, A); soil capacity to store nutrients (SOM, CEC) and soil capacity to supply nutrients (pHH2O, H+Al, Al, Ca, Mg, K, P, BS and SB). For SHI-SMAF, the data set covered was physical (BD, SWSC), chemical (pHH2O, K, P) and biological (OC, MBC, BG) indicators.

### 3.3 Performance of Integrated Indices in Soil Health Assessment

The comparison of soil health indices by using the four strategies showed significant differences (Fig. 3). It is noteworthy that SHI_PCA and SHI_EO exhibit greater variability in their datasets, with lower scores when compared to SHI_SMAF. SHI_SMAF and SHI_FERTBIO showed less variation in scores, promoting better homogenization of the datasets.

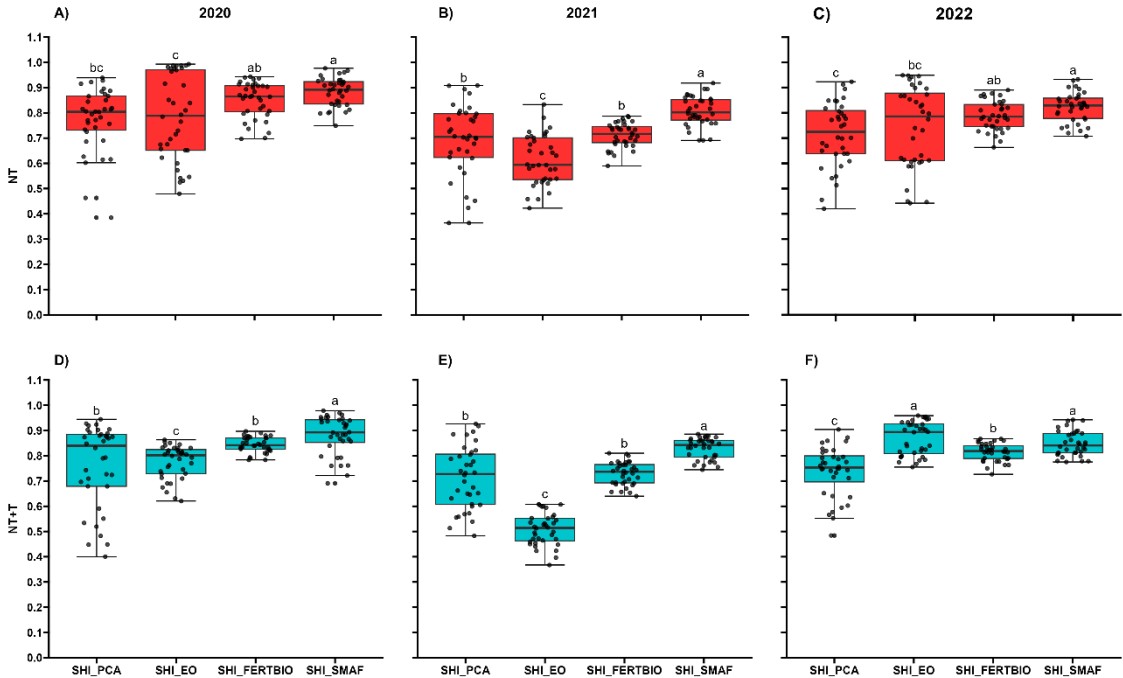

**Figure 3. SMAF consistently showed the highest scores in soil health assessment over the three years of evaluation. Box plots of soil**
**health indices—SHI_PCA, SHI_EO, SHI_FERTIBIO, and SHI_SMAF—measured in 2020 (A and D), 2021 (B and E), and 2022 (C and F) for mega-plots managed under no-till (NT) (A–C) and no-till with terraces (NT+T) (D–F). Different letters indicate statistically significant differences among SHI values within the same year and treatment, according to the Kruskal–Wallis test followed by Dunn test (p < 0.05).**



**3.4 The effect of agricultural terracing on soil health**

Among the four soil health indices (SHI) evaluated, only two—SHI_EO and SHI_FERTIBIO—detected significant differences between the treatments (NT and NT+T) (Fig. 4). SHI_EO showed differences in 2021 and 2022, with a lower score for NT+T (0.51) compared to NT (0.59) in 2021. However, this trend reversed in 2022, when NT+T had a higher score (0.89) than NT (0.79). Similarly, SHI_FERTIBIO indicated a higher value for NT+T (0.82) than for NT (0.78) in 2022. To better understand the variation in SHI performance, the indices were further decomposed into their respective components,

including principal components and physical, chemical, and biological sub-indices.

In the decomposition of SHI_PCA into its principals components, NT+T exhibited higher scores in 2020 for PC4 and PC5, which were primarily represented by TP and MBC (PC4), and by P (PC5). Although MBC showed higher values in NT (Table S3), its eigenvalue was lower than that of TP, meaning TP contributed more strongly to the component score. This resulted in higher overall values in PC4 for NT+T. In 2021, T showed higher scores in PC1, mainly associated with

SWSC—a variable that consistently exhibited higher values under NT+T throughout the study period (Table S3).

The decomposition of SHI_EO into physical, chemical, and biological indicators revealed significant differences between treatments only in the biological index. In 2021, the biological index was lower in NT+T compared to NT, whereas in 2022, it was higher in NT+T. For SHI_FERTIBIO, significant differences in chemical and biological indices were observed only in 2021, with NT+T showing a higher chemical index and a lower biological index than NT.

SHI_SMAF decomposition revealed differences in both 2020 and 2021. In 2020, only the biological index differed, with NT+T showing higher values. However, in 2021, significant differences were observed across all indices: NT+T had higher physical and chemical scores, but a lower biological index compared to NT. This lower biological performance in NT+T in 2021 was consistently detected by three indices—SHI_EO, SHI_FERTIBIO, and SHI_SMAF. Notably, the only biological indicator common to all three indices is BG.







**Figure 4. Soil Health Indices based on expert opinion (SHI_EO) and FERTBIO (SHI_FERTIBIO) distinguished no-till (NT) no-till with terraces (NT+T) plots, primarily due to differences in biological indicators. The graphs display the medians of four Soil Health Indices: SHI_PCA (A–C), SHI_EO (D–F), SHI_FERTIBIO (G–I), and SHI_SMAF (J–L), assessed in 2020 (A, D, G, J), 2021 (B, E, H, K), and 2022 (C, F, I, L) for the NT and NT+T treatments. In addition to the integrated SHI values, the graphs also present their decomposition into respective components, including principal components of SHI_PCA (A–C) and physical, chemical, and biological sub-indices of the other three SHI tested (E–L). Asterisks indicate statistically significant differences between treatment medians according to the Wilcoxon test at the 1% (\*\*\*), 5% (\*\*), and 10% (\*) significance levels; ns: not significant. Medians are based on thirty-six biological replicates.**

## 3.5 Crop yield and relations with soil health indexes

Crop yields under NT and NT+T over the years 2020, 2021, and 2022 are shown in Fig. 5. It can be observed that, in the 2019/20 maize harvest and the 2021 wheat harvest, the NT+T treatment resulted in higher yields compared to NT. For the other harvests, no significant differences were observed between the treatments.

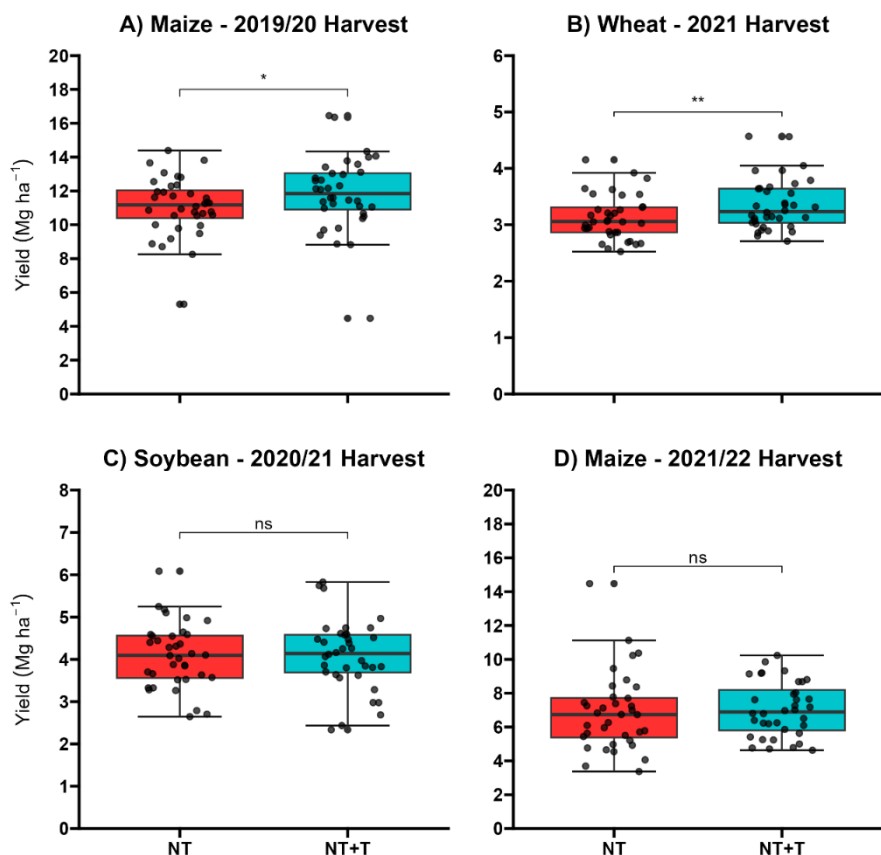

**Figure 5. Maize yield (2019/20 harvest) and wheat yield (2021 harvest) were higher in the no-tillage with terraces (NT+T). The graphs show the median yields of the four harvests that took place in the years 2020, 2021, and 2022: A) Maize - 2019/20 Haverst,**





**B) Wheat – 2021 Haverst, C) Soybean - 2020/21 Harvest and D) Maize - 2021/22 Haverst. Asterisks indicate statistically significant differences between treatment medians according to the Wilcoxon test at the 1% (***), 5% (**), and 10% (*) significance levels; ns: not significant. Medians are based on thirty-six biological replicates.**

Analyzing the SHIs alongside crop productivity over the years 2020, 2021, and 2022 reveals a gradual separation between treatments along the first principal component (Dim1), with the most distinct separation observed in 2022 (Fig. 6). The SHIs show strong correlations with each other. Regarding crop performance, maize productivity in 2020 showed a weaker correlation with the indices. In 2021, wheat exhibited higher productivity under the NT+T, with a stronger association with the SHIs. In 2022, maize productivity was more closely aligned with the indices.




**Figure 6. In 2022, no-till with terraces promoted improvements in soil health, which was reflected in stronger relationships between soil health indices and crop productivity. Principal component analysis (PCA) of soil health indices (SHI_PCA, SHI_EO,**





**SHI_FERTBIO, and SHI_SMAF) and crop yield under no-till (NT) and no-till with terraces (NT+T) for the years 2020 (A), 2021 (B), and 2022 (C). The results are statistically significant at P < 0.05 based on 999 permutations.**

**4 Discussion**

The approaches used to select the indicators that make up the SHIs resulted in the choice of different indicators, which can influence a more or less accurate assessment of the soil health status through the indices. Selection using PCA is widely used in the literature (Marion et al., 2022; Mukherjee and Lal, 2014; Yu et al., 2023), because it allows for the identification and selection of indicators with greater weight and representativeness in the dataset. However, indicators that are more sensitive

to changes in soil health may end up not being chosen (Mukherjee and Lal, 2014; Navas et al., 2011; Rossi et al., 2009).

Expert opinion is a selection strategy that requires more in-depth knowledge of the performance of indicators in soil functions, based on published data and the consensus of the researchers involved. BD and SPR are associated with soil compaction (Reintam et al., 2009) and are easy to determine. The presence of compacted layers in agricultural areas is common, as a result of machine traffic or the natural process of soil particles settling (Hamza; Anderson, 2005). The SWSC

represents the ideal balance between the water-holding capacity of the soil near the surface, recognizing that healthy root development depends on appropriate air and water storage limits in the soil (Reynolds et al., 2002). The chemical indicators (pHCaCl2, P and K) are routinely determined in fertility analyses and are related to soil acidity and soil nutrient availability functions. Organic carbon, representing biological indicators, is a common indicator in studies on soil health (Bünemann et al., 2018), due to its influence on ecosystem processes (Nunes et al., 2021). In southern Brazil, no-till farming is an

important source of carbon for soils (Bernoux et al., 2006; Thomaz; Kurasz, 2023). The BG is a ubiquitous enzyme in the environment, essential for decomposing organic matter and providing energy to soil microorganisms (Deng; Popova, 2011). The enzyme arylsulfatase is studied to investigate the mineralization of sulfur in the soil, as it participates in its cycle in the soil, hydrolyzing sulfate ester bonds and releasing sulfate ions to microorganisms (Acosta-Martínez et al., 2007; Tabatabai; Bremner, 1970).

All the indicators selected for SHI_FERTBIO are linked to the routine analysis of soil fertility and functions and are aimed at proper management of fertilizer nutrients for soil health (Mendes et al., 2021b). However, the absence of indicators related to soil physical properties is concerning, as this may lead to a partial or inaccurate assessment of soil health (Yu et al., 2023; Bünemann et al., 2018). The authors acknowledge this absence but emphasize that biological indicators (OC, BG, and A) have a significant influence on physical properties. For SHI_SMAF, the selection of indicators focuses on those associated

with the total or partial performance of soil functions and includes only those for which scoring curves have been developed and validated within the SMAF protocol (Andrews et al., 2004).

The differences in variability observed among the SHIs are directly related to the strategies used for their construction (Fig. 2). SHI_PCA and SHI_EO showed greater variability in the scores of their components. According to Jolliffe and Cadima (2016), PCA aims to reduce the dimensionality of a given dataset while preserving maximum variability. Therefore, the





indices generated from the indicators selected through PCA are directly conditioned by the natural variability that occurs within the physical environment. Therefore, PCA may assign greater weight to the most sensitive indicators in the dataset, which does not necessarily imply a stronger relationship between these indicators and soil functioning (Rinot et al., 2019). SHI_EO, in turn, had its indicator weights assigned using the covariance matrix, which is based on the covariance between two variables within a dataset. Thus, variables with greater representativeness in the dataset will have higher covariance

(Jolliffe, 2002). In other words, there should be a correlation between the variable and the final index to assess how much each variable influences the final index. In the SHI_FERTBIO and SHI_SMAF methods, the indicators are weighted to average values, with the adjustment obtained through pre-established proportions that are independent of the evaluated indicators' conditions (their natural variability). According to Marion et al. (2022), the weighting and standardization of the dataset maintain uniformity regardless of the study location or system. Thus, the PCA and EO indicators are more sensitive

to local and environmental variations and, consequently, exhibit a greater range than the others.

In 2021, SHI_PCA, SHI_FERTIBIO, and SHI_SMAF showed significant differences in their decomposed indices, although these differences were not reflected in the integrated SHI values (Fig. 3). Conversely, in 2022, SHI_FERTIBIO displayed differences in the integrated SHI, but no significant variation was observed in its decomposed components. The only index that consistently captured differences between treatments—both in its integrated and decomposed forms—was SHI_EO. In

the studies by Cherubin et al. (2016) and (Marion et al. (2022), the strategy for determining SHI based on expert opinion proved effective in detecting differences between treatments, even with a reduced number of indicators. The authors emphasize that opting for simpler strategies may be more appropriate for decision-making, as it does not require a large dataset, as is the case with SHI_PCA. Martín-Sanz et al. (2022) found that the index generated by the indicators selected by PCA was unable to identify differences in soil health.

Overall, the biological index was the most sensitive in detecting treatment differences. Of the 12 significant differences identified across all specific indices, 6 were biological, while 3 were chemical and 3 physicals. At the stage of determining the specific indices (Table 4) for the entire period evaluated, the biological indicators had already shown greater sensitivity in differentiating the effects of the treatments. Biological indicators are dynamic and more susceptible to environmental changes. In this way, they can anticipate any disturbance to the sustainability of the environment at an early stage (Cardoso

et al., 2013; Masto et al., 2008).

Awareness of how changes in soil health affect crop productivity leads to efficient input use and increased agricultural production yield (Hemmati et al., 2022). Productivity is a critical plant biological indicator for sustainable management of agricultural areas. In this study, corn productivity in the 2019/20 season and wheat in the 2021 season were higher in the no-till with terraces system (Fig. 5). However, in 2020, soil health indices did not yet clearly differentiate between treatments, as

it was the first year of evaluation post-homogenization of the area with subsoiling. Nevertheless, by 2022, the indices showed a better correlation with each other and with productivity, as well as indicating improved soil health for the no-till with terraces (Fig. 6), reinforcing the consistency of approaches and the influence of soil health on crop performance, with agricultural terracing enhancing soil health. However, it must be acknowledged that the relationship between productivity

and soil health can vary annually, influenced by factors such as climate, management practices, and crop type (Walder et al.,
2023). This study underscores the importance of long-term experiments in areas with agricultural terracing to understand
their improvements in soil health and agricultural productivity.

## 5 Conclusions

Four approaches of soil health were evaluated in no-till with and without agricultural terraces, and variations among the tests
were observed. The SHI-SMAF and SHI-PCA were not sensitive, while the SHI-EO and SHI-FERTBIO showed differences
between treatments. In this case, agricultural terracing improved soil functioning in the final year of evaluation. This effect is
mainly associated with biological indicators of soil health.

No-till with terraces resulted in higher maize productivity in the 2019/20 haverst and higher wheat productivity in the 2021
season. Crop productivity was associated with soil health indices, showing gradual improvements over the years in this
management system.

## Funding sources

To the Foundation Araucária for funding the project (Agreement FA 302/2024). To FAEP, SENAR-PR, and Paraná Network
of Agricultural Research and Applied Training for the support.

## Acknowledgements

To Dr. Ieda Carvalho Mendes for her help in obtaining the SHI_FERTBIO. I acknowledge the use of ChatGPT
[https://chat.openai.com/] to translate the text into English.

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
