# Peer review of "Soil health approaches to assess the impacts of no-tillage with agricultural terraces in southern Brazil"

_EGUsphere, 2025_

## Author Comment (AC1)

Respected Sir,

We sincerely appreciate the detailed and constructive feedback provided by the reviewer. Below, we present our point-by-point responses, indicating how each suggestion was incorporated into the revised version.

**GENERAL COMMENTS**

In response, we have revised the Introduction to:

1. Clarify the scientific gap, explicitly stating that previous studies have not directly compared the sensitivity of different soil health assessment methods in detecting short-term effects of conservation practices such as terracing in subtropical no-till systems.
2. Emphasize the conceptual contribution of the study, highlighting that the novelty lies in the comparative evaluation of the diagnostic capacity of these approaches in a highly relevant subtropical agroecosystem.

**SPECIFIC COMMENTS**

**Novelty: The introduction should more clearly state the specific knowledge gap: a direct, comparative evaluation of the sensitivity of these established methods for detecting short-term impacts of conservation practices like terracing.**

This comment has been fully accepted. We have revised the end of the Introduction to explicitly address the suggestion.

"Although terracing in no-till systems is widely recognized for reducing erosion and improving water infiltration (Fuentes-Guevara et al., 2024; Lal, 2020; Panagos et al., 2015), their effects on soil health under subtropical field conditions remain poorly understood, despite the availability of multiple assessment methods. Moreover, no study has directly compared the sensitivity of different soil health approaches in detecting short-term impacts of terracing in subtropical no-till systems."

**Methods Clarification – Expert Opinion (EO): The description of the Expert Opinion (EO) method requires clarification. Specify if the initial indicator selection was purely based on expertise, while only the weighting was data-derived, to avoid a circular argument.**

We reformulated

"In the EO approach, indicator selection was guided solely by prior technical expertise and evidence from the scientific literature concerning the sensitivity and functional relevance of physical, chemical, and biological soil indicators. No data-driven criteria were applied in this selection step. This selection prioritizes the indicators that are most sensitive to variations in soil function and those that are easier to determine in the field or laboratory and to interpret."

"The SHI_EO was integrated using the covariance matrix weights, in a manner based on the approaches described by Manly (2008) and Jolliffe (2002) for different multivariate analysis methods. The SHI_EO integration method consisted of an analysis of the proportional contribution of variables to a composite index, ensuring that the integration step reflects the variability observed in the dataset while keeping the selection step independent of the data."

**Conclusions**

We completely agree. The conclusion has been reformulated as follows:

"Among the approaches tested, SHI_EO and SHI_FERTBIO demonstrated greater sensitivity in detecting short-term changes in soil health."

**TECHNICAL CORRECTIONS**

All line numbers mentioned in this response correspond to the revised manuscript.

L 78:

Correction made: "their effects on soil health".

Throughout the manuscript, the word "Havest" was corrected to "Harvest".

---

## Author Comment (AC2)

Respected Sir,

We sincerely thank the reviewer for the careful reading and constructive suggestions. We fully agree that the manuscript required a clearer and more cohesive narrative regarding its primary objective.

Following the reviewer's recommendation, we have now explicitly defined the main objective of the study as the comparative evaluation of the sensitivity and performance of four soil health assessment approaches, using agricultural terracing under no-till as an applied case study.

Accordingly, we revised the Introduction, Discussion, and Conclusions to consistently reflect this focus. The discussion was expanded to address (i) the physical, chemical, and biological drivers underlying the temporal variability between NT and NT+T, and (ii) the practical applicability, scalability, and potential transferability of these approaches to other soil types and regions.

We believe these changes have substantially improved the clarity, coherence, and impact of the manuscript.

**SPECIFIC COMMENTS**

**P2, L49: Please define 'FERTBIO'. Is this an acronym or a specific program? Please cite the developer or governing body.**

In the Introduction, we have added the definition of FERTBIO as the Soil Fertility and Biology sampling concept, embedded within Embrapa's Soil Bioanalysis Technology (SoilBio) framework. The developing and governing institution (Embrapa), as well as the key references associated with the development and application of this approach, are now explicitly indicated in the manuscript (Mendes et al., 2021a, 2021b, 2024).

**P2, L60: Please specify the crop rotation or specific crops used in the no-till system.**

This information has been added to the Materials and Methods section.

**P3, L65: Please clarify the term "tropical agriculture."**

We thank the reviewer for this important observation. The expression "tropical agriculture" has been removed from the manuscript, and the text was revised to provide

a more precise geographic and environmental context, now referring to agricultural systems in southern Brazil and to subtropical field conditions, thereby eliminating any climatic ambiguity.

"Given Brazil's leadership in agriculture and the widespread adoption of no-tillage, which is applied to 33 million hectares of the country's cropland (Fuentes-Llanillo et al., 2021), but which alone is not sufficient to control soil loss on sloping lands (Merten et al. 2015), soil erosion remains an ongoing challenge, especially in areas with rugged terrain. Conservation techniques such as no-till and terracing have been essential strategies to mitigate soil degradation and improve agricultural sustainability in the region (Merten et al., 2015).

Although terracing in no-till systems is widely recognized for reducing erosion and improving water infiltration (Fuentes-Guevara et al., 2024; Lal, 2020; Panagos et al., 2015), their effects on soil health under subtropical field conditions remain poorly understood, despite the availability of multiple assessment methods. Moreover, no study has directly compared the sensitivity of different soil health approaches in detecting short-term impacts of terracing in subtropical no-till systems. Therefore, this study aims to comparatively evaluate the performance and sensitivity of four integrative soil health assessment approaches in detecting short-term changes, using agricultural terracing under no-till management as a field-based case study in southern Brazil, and to identify the most responsive soil health indicators for monitoring these systems."

**P3, L71: Define the SHI abbreviation at its first mention (e.g., Soil Health Indicators).**

The abbreviation was defined at its first occurrence in the Introduction.

"...generate integrated soil health indices (SHIs), allowing for a holistic..."

**P4, L95: Please state whether any irrigation was implemented or if the study was strictly rain-fed.**

We have added:

"The experiment was conducted entirely under rainfed conditions throughout the entire study period."

**P4, L99: Clarify if "field capacity" refers specifically to volumetric water content at field capacity.**

This has now been clarified in the Materials and Methods section. The manuscript now explicitly defines field capacity as the volumetric water content at field capacity.

**P5, L105: I note SHI is defined here as "Soil Health Indices." As mentioned above, please ensure this is defined at the first usage in Line 71.**

This has now been corrected. The abbreviation SHIs (Soil Health Indices) is now defined at its first occurrence in the Introduction, and the redundant definition has been removed accordingly.

**P5, L120: Please correct the citation formatting: "...proposed by Andrews et al. (2004), Stott et al. (2010), and Wienhold et al. (2009) were selected."**

We have reformulated the text as follows: "[...] proposed by Andrews et al. (2004), Stott et al. (2010), and Wienhold et al. (2009) were selected."

**P7, L134: In the table, please define what "A" represents in the context of your data categories.**

This has been corrected. The manuscript now explicitly defines "A" as arylsulfatase activity in the Table 2 legend.